# WWOX Controls Cell Survival, Immune Response and Disease Progression by pY33 to pS14 Transition to Alternate Signaling Partners

**DOI:** 10.3390/cells11142137

**Published:** 2022-07-07

**Authors:** Tsung-Yun Liu, Ganesan Nagarajan, Ming-Fu Chiang, Shenq-Shyang Huang, Tzu-Chia Lin, Yu-An Chen, Chun-I Sze, Nan-Shan Chang

**Affiliations:** 1Laboratory of Molecular Immunology, Institute of Molecular Medicine, College of Medicine, National Cheng Kung University, Tainan 70101, Taiwan; s91009s91008@gmail.com (T.-Y.L.); nagarajangi@mail.com (G.N.); louis19862002@gmail.com (S.-S.H.); cutebaby9901053@gmail.com (T.-C.L.); momofish0716@gmail.com (Y.-A.C.); 2Department of Basic Sciences, PYD, King Faisal University, Al Hofuf 36362, Saudi Arabia; 3Graduate Institute of Injury Prevention and Control, Taipei Medical University, Taipei 11031, Taiwan; chiang66@gmail.com; 4Anesthesiology Department, National Cheng Kung University, Tainan 70101, Taiwan; 5Department of Cell Biology and Anatomy, National Cheng Kung University, Tainan 70101, Taiwan; chuni.sze@gmail.com; 6Graduate Institute of Biomedical Sciences, College of Medicine, China Medical University, Taichung 40402, Taiwan; 7Department of Neurochemistry, New York State Institute for Basic Research in Developmental Disabilities, New York, NY 10314, USA

**Keywords:** tumor suppressor, p53, WWOX, TIAF1, binding proteins, cancer, Alzheimer’s disease, therapeutic peptides

## Abstract

Tumor suppressor WWOX inhibits cancer growth and retards Alzheimer’s disease (AD) progression. Supporting evidence shows that the more strongly WWOX binds intracellular protein partners, the weaker is cancer cell growth in vivo. Whether this correlates with retardation of AD progression is unknown. Two functional forms of WWOX exhibit opposite functions. pY33-WWOX is proapoptotic and anticancer, and is essential for maintaining normal physiology. In contrast, pS14-WWOX is accumulated in the lesions of cancers and AD brains, and suppression of WWOX phosphorylation at S14 by a short peptide Zfra abolishes cancer growth and retardation of AD progression. In parallel, synthetic Zfra4-10 or WWOX7-21 peptide strengthens the binding of endogenous WWOX with intracellular protein partners leading to cancer suppression. Indeed, Zfra4-10 is potent in restoring memory loss in triple transgenic mice for AD (3xTg) by blocking the aggregation of amyloid beta 42 (Aβ42), enhancing degradation of aggregated proteins, and inhibiting activation of inflammatory NF-κB. In light of the findings, Zfra4-10-mediated suppression of cancer and AD is due, in part, to an enhanced binding of endogenous WWOX and its binding partners. In this perspective review article, we detail the molecular action of WWOX in the HYAL-2/WWOX/SMAD4 signaling for biological effects, and discuss WWOX phosphorylation forms in interacting with binding partners, leading to suppression of cancer growth and retardation of AD progression.

## 1. Introduction

The protein-protein interaction network supports normal physiology for cell survival and death [1,2,3]. Alteration of a single component or multiple components in the signaling network may result in enhanced or reduced cancer cell survival. That is, the flow of the altered signaling pathway is redirected or shut down unexpectedly, or runs uncontrollably. The strength of protein-protein binding interactions during signaling in cells is affected by environmental factors such as local ionic strength and pH, amino acid compositions in the binding motif(s) or domain(s), competitors in the protein-binding cascade, and transient changes in the protein conformation upon interacting with another binding partner [1,2,3]. Under physiological conditions, a signaling pathway is initiated by an extracellular trigger or a ligand to allow the normal operation of protein-protein binding and progression of the signaling cascade. However, the process by which the protein-protein signaling is redirected from physiological to pathological conditions is largely unknown.

## 2. Protein Interaction Network in Normal Signaling and Diseases

Computational analysis may provide the prediction of the protein-protein interactions that link to cancer progression. Gene expression analysis from thousands of cancer samples at the Cancer Genome Atlas yields differential arrays or clusters of protein-protein signaling network among cancers. These aberrant signal pathways are specific for the progression of one or multiple cancer types [1,2,3]. Nonetheless, empirical determination for the dynamic protein-protein binding and dissociation in cell levels is needed, so as to build the big picture. For example, formation of an intracellular large protein complex may involve changes in protein conformations, enzymatic cleaves, energy requirement, heat release, and interactions with small molecules, ions, and/or amino acids [4,5,6,7,8,9,10].

In this review article, we will mainly detail the extracellular stimuli-mediated activation of WW domain-containing oxidoreductase (WWOX) signaling network. We will discuss how transforming growth factor beta (TGF-β1) and hyaluronan activate WWOX and downstream normal or aberrant protein partners and the potential consequences for the progression of cancer and Alzheimer’s disease (AD) (Figure 1). In principle, we will address in a stepwise manner how WWOX signals interact with p53, HYAL-2 and SMAD4 to generate significant biological events. Our writing flow is as follows: (1) WWOX structure and domains involved in protein binding, controlling cell adhesion and migration, cell-cell recognition, and relevance to cancer metastasis and neuronal heterotopia, (2) reduction in Y33 phosphorylation and subsequent upregulation of S14 phosphorylation in WWOX during disease progression, (3) measuring WWOX signaling by real-time Förster resonance energy transfer (FRET) microscopy, (4) TGF-β1-regulated activation of WWOX and binding proteins (e.g., p53 and TIAF1) for cell death by time-lapse microscopy, and (5) competitive binding interactions leading to altered signaling. The following scheme illustrates the flow of this review article.

## 3. WW Domain-Containing Oxidoreductase (WWOX)

WWOX and its binding proteins form a complicated signaling network which is needed for normal physiology and/or disease progression [11,12,13,14,15,16,17,18,19,20,21]. WWOX was first discovered in 2000 [12]. Most recently, the human *WWOX* gene has been defined as a risk factor for AD [22]. Currently, there are many outstanding review articles documented in the PubMed database. In brief, WWOX protein possesses two *N*-terminal WW domains, a *C*-terminal short chain alcohol dehydrogenase/reductase (SDR) domain, and a nuclear localization signal present in between the WW domains [12] (Figure 2A). A mitochondria-targeting region is located in the SDR domain. Both WW and SDR domains have their own specific protein-binding partners, as reviewed in a recent article [19]. The first WW domain (WW1) possesses two tryptophans and binds PPXY (e.g., PPPY) or LPXY motif [12,23] motif in the target proteins, where X is any amino acid. However, when Y33 in the WW1 becomes phosphorylated, pY33-WWOX has an expanded binding capability with numerous proteins.

Overall, WWOX participates in many ligand/receptor-mediated signaling pathways such as WNT/β-catenin [24], JNK [25], TGFβ1/HYAL-2 [26], TGFβ/SMADs [26,27], HIPPO [28], HCG/MET [29], ], JAK2/STAT3 [30,31,32], p73 [33], and others. WW2 domain has only one tryptophan and its function is largely unknown. WW2 is proposed to line up with WW1 in facilitating the binding of WWOX with its interacting partners [34]. The SDR domain has an NSYK (Asn-Ser-Tyr-Lys) motif, which binds sex steroid hormones androgen and estrogen [12]. The SDR domain also exhibits redox activity [35,36,37], and binds tau and GSK-3β to prevent tau hyperphosphorylation [19,38,39,40]. Whether the redox activity of SDR domain limits GSK-3β-mediated tau hyperphosphorylation and subsequent tau aggregation is unknown [38,40].

## 4. WWOX Controls Cell Migration, Cell-Cell Recognition, and Neuronal Heterotopia

Of utmost concern is that human newborns with *WWOX* gene deficiency suffer severe neural diseases such as seizure, encephalopathy, and early death [19,20,21,41,42]. There is no cure for the disease. Loss of WWOX enhances cell mobility [30,36]. *WWOX* gene deficiency in newborns results in neuronal migration disorders (or neuronal heterotopia) that lead to epileptic seizures [40,42]. Two types of cells based upon their expression of functional or dysfunctional WWOX have recently been identified [36,38]. Cells which are deficient in WWOX protein, or express dysfunctional WWOX, can be considered as metastatic cancer cells (designated as WWOXd).

At room temperature, WWOXd cells are less efficient in generating Ca^2+^ influx and undergo non-apoptotic explosion in response to UV irradiation. In contrast, functional WWOX-expressing cells (designated WWOXf) exhibit non-apoptotic nucleus-dependent bubbling cell death (BCD) [17] and efficient Ca^2+^ influx caused by UV or apoptotic stress at room temperature [36]. Formation of a nitric oxide (NO)-containing nuclear bubble per cell during BCD is due to UV-induced upregulation of NO synthase 2 (NOS2) [17]. WWOXf cells, which are mainly normal and benign cancer cells, migrate collectively and force the individually migrating WWOXd cells to undergo retrograde migration [36,37] (Figure 2B,C). WWOXd cells, in return, induce WWOXf cells to undergo apoptosis from a distance without physical contact. The cytokines responsible for the induced apoptosis are unknown [36,37]. During cell-to-cell encounter, WWOXd cells exhibit activation of MIF, HYAL-2, Eph, and Wnt pathways, which converge to the MEK/ERK signaling and enables the cells to move away from WWOXf cells (Figure 2B,C). Specific antibodies against MIF or HYAL-2, or inhibitors for WNT, cause WWOXf to greet WWOXd cells, and both cells merge eventually (Figure 2B,C). Metastatic cancer cell-derived TGF-β1 allows merger of WWOXf with WWOXd cells [36,37]. A detailed signaling for the pathway of WNT, HYAL-2, EPH-Ephrin, EGF/EFGR, or MIF linking to ERK (or ERK1/2) is shown (Figure 3). The CD44/HYAL-2 complex prevents the binding of TGF-β1 with HYAL-2. CD44 does not appear to be involved in the HYAL-2/WWOX/SMAD4 signaling [18] (Figure 3). Together, these observations suggest WWOX controls cell migration and cell-to-cell recognition and plays a crucial role in neuronal heterotopia that contributes to epileptic seizure [30,35,36,37].

## 5. WWOX Signaling Network

By STRING analysis (https://string-db.org/cgi/network?taskId=brQHTnXWILzL&sessionId=b73ctYMesgwG) (accessed on 19 June 2022), the first level of WWOX-binding protein network reveals that WWOX has connection with 5 interactors, namely TP53, ERBB4, DVL2, FAM189B, and WBP2 [11,25,43,44,45,46,47,48]. The binding has been validated by experimental approaches (Figure 4). Further analysis expanding the interactors up to 26 and 136, respectively, shows many biological process and molecular functions, including beta-catenin destruction complex assembly, response to metformin, ezrin [49], and many others. Again, the network can be expanded tremendously. The significance of this approach is that the data show the high complexity in the biological network and yet precise generation of functional machinery. The downside is that the analysis does not show domain/domain-specific binding interactions. Binding affinity-based interaction network is not clear. The computational approach also fails to prove our empirical observations that the stronger the binding of WWOX with intracellular proteins, the better the suppression of cancer growth and retardation of AD progression [19].

## 6. pY33 to pS14 Transition in WWOX during Disease Progression

WWOX was originally considered as a tumor suppressor. pY33-WWOX is believed to maintain normal physiology and suppress pathological progression [19,38]. For example, transiently overexpressed pY33-WWOX induces cancer cell death and causes death of damaged neurons [12,43,44]. pY33-WWOX complexes with endogenous p-cJUN, p-CREB, and NF-κB p65 to undergo nuclear translocation for blocking or inducing death of damaged neurons [43]. Moreover, pY33-WWOX physically binds JNK1 and ERK and blocks the enzyme-mediated tau hyperphosphorylation [19,38]. Thus, WWOXf cells need pY33-WWOX to exert normal functions.

Under pathological conditions, proapoptotic pY33-WWOX is downregulated and shifted to pS14-WWOX. pS14-WWOX is accumulated in the lesions of cancer and AD brains [19,21,50,51,52,53,54]. Suppression of pS14-WWOX by a small peptide Zfra (zinc finger-like protein that regulates apoptosis) leads to inhibition of cancer growth and blocking of AD progression [50,51], suggesting that WWOX is not an authentic tumor suppressor. Metastatic murine breast 4T1 cells, a WWOXd cell line, tend to exhibit spheres and pS14-WWOX overexpression is mainly in the spheres [36,54]. In contrast, pY33-WWOX expression is low [54]. Chemotherapeutic drug ceritinib-mediated suppression of 4T1 cell growth is associated with pS14-WWOX downregulation [54]. Presumably, cells expressing pS14-WWOX can be considered as a WWOXd phenotype.

## 7. WWOX Functional Measurement by Time-Lapse FRET Microscopy

Förster resonance energy transfer (FRET) microscopy is a feasible approach to measure WWOX function in a real-time manner. FRET can be utilized to determine spatial proximity among proteins either at a single or multiple protein levels in cultured cells in a real-time mode [18,43,44,55,56,57,58,59,60,61,62]. For example, binding of a CFP (cyan fluorescence protein)-tagged bait (or donor) protein with a YFP (yellow fluorescence protein)-tagged target (or acceptor) protein results in energy flow from the donor to the acceptor. In other words, an excitation wavelength is used to excite a donor protein (e.g., CFP tagged), and once excited the donor protein transfers the resulting energy to the acceptor (e.g., YFP tagged), which subsequently emits a longer wavelength with a lower energy. To be effective in signal transduction, protein proximity at 1–10 nm is needed for FRET detection. Furthermore, energy release from the first donor protein can excite two acceptors tagged with different fluorophores for determining parallel signaling pathways. We have first developed the technology for energy transfer from the first excited donor protein going directly to the second acceptor, and then the emitted energy from the second acceptor going to the third one [56]. This approach allows us to follow the signaling flow, the eventual outcome, and novel kinetics of protein binding [43,56,58]. We believe that the technology for signaling more than 3 partners in a pathway can be developed by tagging each protein with a small fluorescent probe. This is to prevent the self-aggregation of GFP or related fluorescent proteins.

## 8. TGF-β1 Induction of Initial Driving Force and Then Execution Force for Protein-Protein Binding and Cell Death: TIAF1 Is a Blocker of TGF-β1/SMAD Signaling 

Analysis of protein-protein binding kinetics by time-lapse FRET microscopy reveals the time-related changes in the binding force for a single protein or two to three binding protein partners, as well as alterations in cell morphology [44,56,58]. Transforming growth factor beta 1 (TGF-β1) induces the time-dependent self-polymerization of TIAF1 (TGF-β-induced antiapoptotic factor) in colon HCT116 cells [63,64,65]. Indeed, TGF-β induces TIAF1 self-aggregation via type II receptor-independent signaling that leads to generation of amyloid β plaques in Alzheimer’s disease [63]. HCT116 is a WWOXf cell line [36]. WWOX binds TIAF1 to prevent the protein from undergoing self-polymerization [64,65,66]. TIAF1 aggregates can be found in the lesions of AD brains [50]. When TGF-β1-induced TIAF1 self-polymerization (using ECFP-TIAF1 and EYFP-TIAF1) reaches a maximal extent, cells start to undergo apoptosis [63,64,65]. Indeed, TGF-β1 stimulates aggregation of transiently overexpressed EGFP-TIAF1 to form intracellular green punctate prior to membrane blebbing and apoptosis [63]. When an equal amount of cDNA expression constructs for ECFP-SMAD4 and EYFP-TIAF1 is co-expressed in HCT116 cells, binding of SMAD4 with TIAF1 does not occur. No TGF-β1-induced aggregation of TIAF1 and apoptosis is observed [63], suggesting that SMAD4 prevents TIAF1 self-aggregation.

In contrast, when non-small cell lung cancer NCI-H1299 cells express a lower amount of ectopic EYFP -TIAF1 and a higher amount of ECFP-SMAD4, TGF-β1 increases the binding of EYFP-TIAF1 with ECFP-SMAD4, as determined by time-lapse FRET microscopy [63], NCI-H1299 is also a WWOXf cell line [36], Notably, an initial binding force between TIAF1 with SMAD4 is gradually increased followed by reduction, hereby designated as phase I. In phase II, the binding force between TIAF1 and SMAD4 becomes stronger than phase I by 30%. Meanwhile, there is a sharp decrease in cell volumes, indicating cells undergo apoptosis [63], In comparison, when cells express a greater amount of EYFP-TIAF1 than ECFP-SMAD4, TGF-β1-induced binding force for EYFP-TIAF1 and ECFP-SMAD4 is reduced in phases I and II. The extent of cell shrinkage and death is also retarded. While transiently overexpressed TIAF1 strongly binds endogenous SMAD2, 3 and 4 [35,63,64,65,66], TIAF1 blocks the TGF-β1/SMAD signaling.

Overall, during signaling transduction, an initial force is needed for protein-protein binding in order to drive the signal pathway forward. Importantly, protein concentrations affect the binding status for two proteins, yielding distinct biological effects such as signaling moving forward or getting on hold. Furthermore, despite both HCT116 and NCI-H1299 cells possessing functional WWOX, their subcellular signaling pathways are likely to be different. Thus, under SMAD4 and TIAF1 overexpression, the biological outcome is expected to be different.

## 9. The Dynamics of WWOX/TIAF1/p53 Triad Formation and Functional Antagonism between p53 and WWOX for Enhancing the Progression of Cancer and Alzheimer’s Disease

How WWOX deficiency (or downregulation) or pS14-WWOX upregulation contributes to disease progression is largely unknown [50,51,67]. Supporting evidence shows that the status of WWOX binding with p53 and TIAF1 may play a role in cancer and AD progression. Intracellular p53 and WWOX may counteract each other functionally, and thereby lead to cancer growth enhancement and development of AD pathologies in vivo [35]. WWOX physically binds and stabilizes wild type p53 from being degraded by the proteasomal system [48]. Under stress conditions, tumor suppressors p53 and WWOX form a complex with TIAF1, and the WWOX/TIAF1/p53 triad strongly inhibits cancer cell growth, migration, anchorage-independent transforming growth, and SMAD promoter activation, and ultimately causes cancer cell apoptosis [35]. The WWOX/TIAF1/p53 triad has been confirmed by co-immunoprecipitation and FRET microscopy.

Phosphorylated pY33-WWOX binds pS46-p53, and TIAF1 binds pY33-WWOX [35,48]. Notably, TIAF1 binds only activated p53 (pS46-p53) but not non-activated p53 [35]. Transiently overexpressed TIAF1 undergoes polymerization and blocks nuclear translocation of cytosolic p53 and WWOX. When activated p53 and TIAF1 complex together in the absence of WWOX, the complex strongly suppresses anchorage-independent growth [35]. p53/TIAF1 inhibits SMAD promoter activation as regulated by WWOX or SMAD4 [39,52].

There are 12 p53 isoforms [68]. Among these, Δ133p53γ isoform is the strongest in suppressing cancer cell migration, and this positively correlates with Δ133p53γ-mediated SMAD promoter activation [35]. We do not exclude the possibility that Δ133p53γ undergoes self-association and this provides a driving energy to cause SMAD promoter activation and inhibition of cell migration. Unlike the full-length p53, Δ133p53 isoforms α, β, and γ do not have the transactivation domains and the beginning of the DNA-binding domain [68,69]. Additionally, Δ133p53 isoforms play a critical role in cancer, aging, neurodegeneration and immunity [68,69].

Most interestingly, ectopic WWOX inhibits lung cancer NCI-H1299-mediated inflammatory splenomegaly and cancer cell growth in nude mice, and that p53 counteracts the effect of WWOX in these mice. When mice have ongoing growth of p53/WWOX-expressing lung cancer cells, the mice tend to have BACE (β-secretase 1) upregulation, APP (amyloid precursor protein) degradation, tau tangle formation, and amyloid β generation in the brain and lung [35]. That is, functional antagonism between p53 and WWOX leads to enhanced cancer growth and accelerated neurodegeneration in vivo [35]. We do not exclude the possibility that p53/TIAF1/WWOX triad becomes aggregated in the brain and contributes to aggregation of tau and amyloid beta of the AD pathologies.

Whether the p53/TIAF1/WWOX triad induces cancer cell death in vivo is unknown. When phosphorylation of WWOX in Y33 is downregulated and S14 upregulated, the binding strength between WWOX and p53 is expected to be reduced. This facilitates the growth of cancer and the progression of Alzheimer’s disease [50,51]. By the same token, binding of pS14-WWOX with other intracellular proteins is weakened and thereby favors the growth of cancer and the progression of AD [19,53]. In parallel, when WWOX is downregulated or Y33 phosphorylation is blocked, UV-induced p53 protein expression is suppressed, even though p53 mRNA levels are stable [47]. Thus, the proteomic profiles of intracellular protein-binding partners for pS14- and pY33-WWOX, respectively, remains to be established.

## 10. Identification of HYAL-2/WWOX/SMAD4 Signaling in Regulating Physiological and Pathological Events

We have established the HYAL-2/WWOX/SMAD4 signaling, which controls cell proliferation, differentiation, and death [18,26,44,50,51,52,53]. The HYAL-2/WWOX/SMAD4 complex plays a critical role in hyaluronan or TGF-β1-mediated cell proliferation or apoptosis [26,44], neuronal death during traumatic brain injury in vivo [15,18,19,44], and activation of cytotoxic HYAL-2+ CD3− CD19− Z lymphocytes in vivo [50,51,52,53,54]. Both TGF-β1 and hyaluronan competitively bind membrane HYAL-2 to drive the signaling with WWOX and SMAD4 [26,44], whereas the competition-related biological outcome is unknown.

The HYAL-2/WWOX/SMAD4 complex has been confirmed by co-immunoprecipitation, immunoelectron microscopy, and time-lapse FRET microscopy [26,44,50,51,52,53,54,70]. By cytoplasmic Ras-based yeast two-hybrid analysis [11,25,26,38,48,70,71,72], SMAD4 is tagged with a membrane-targeting signal to allow the protein to anchor on the yeast cell membrane for binding with cytosolic baits (Figure 4A). Many WWOX cDNA constructs are utilized for cytosolic expression in the yeast, including full-length, specific domains, Y33R mutant, and other specified WWOX proteins [44]. These proteins are expressed in the cytoplasm of yeast as baits. When wild-type WWOX physically binds the membrane-bound SMAD4 via its *N*-terminal first WW domain (WW1) to activate the Ras signal pathway, yeast cells start to grow at 37 °C in a selective agarose plate containing galactose. Without binding, yeast cells cannot grow at 37 °C. The binding is Y33-phosphorylation dependent. Alteration of Y33 to R33, i.e., WW1 (Y33R), results in abrogation of binding of WW1 (Y33R) to membrane SMAD4. Additionally, SMAD4 physically interacts with the *C*-terminal SDR domain of WWOX [44]. Under similar conditions, WW1 binds HYAL-2, which is Y33-phosphorylation dependent [44]. That is, WW1(Y33R) cannot bind HYAL-2. Full-length HYAL-2 does not bind the full-length SMAD4, indicating that WWOX connects the binding of HYAL-2 with SMAD4. Thus, both HYAL-2 and SMAD4 bind the *N*-terminal pY33-WW1 in WWOX, and SMAD4 also binds the *C*-terminal SDR domain (Figure 5A).

HYAL-2 and SMAD4 competitively interact with the WW1 domain of WWOX [64] (Figure 5B). Increasing amounts of ectopic SMAD4 block the binding between WWOX and HYAL-2, and thereby inhibit the yeast cell growth (Figure 5C). We reported that WW domain physically binds SDR domain in WWOX as a non-activated form, and WW or SDR domain can undergo self-polymerization as determined by FRET microscopy [36]. In response to TGF-β1, UV, hyaluronan, or other stress stimuli, folded WWOX opens up and becomes Y33 phosphorylated as an activated form to interact with HYAL-2 and SMAD4 [26,44] (Figure 5C). Similarly, activated pY33-WWOX strongly binds pS46-p53 to induce cell death in the nucleus [48].

## 11. A WWOX7-21 Epitope Peptide Drives the HYAL-2/WWOX/SMAD4 Signaling

The mechanisms by which WWOX-mediated cancer suppression and inhibition of neurodegeneration take place are largely unknown. There are two surface exposed epitopes in WWOX, which are at amino acids #7-21 and #286-299 [36,53,54]. Synthetic WWOX7-21 peptide, or truncation down to 5-amino acid WWOX7-11, strongly blocks and prevents the growth and metastasis of melanoma and skin cancer cells in mice [54]. WWOX286-299 also inhibits cancer cell growth, whereas it fails to block cancer metastasis to the lung and liver [54].

By time-lapse microscopy, we determined that antibody against WWOX7-21 suppresses ceritinib-mediated breast 4T1 cell sphere explosion and death (Figure 6A,C; Appendix A). 4T1 cell spheres express many makers of stem cells (e.g., Sox2, Oct4 and Nanog). Ceritinib is an antineoplastic kinase inhibitor for treating anaplastic lymphoma kinase (ALK)-positive metastatic non-small cell lung cancer (NSCLC) [73]. In contrast, WWOX7-21 peptide potently enhances the function of ceritinib in causing the explosion and death of 4T1 cell spheres (Figure 6B,C) [54]. pS14-WWOX7-21 peptide or WWOX7-21 antibody supports cancer survival by blocking ceritinib cytotoxicity (Figure 6A–C). Further analysis reveals that ceritinib-mediated cell death is due to rapid upregulation of proapoptotic pY33-WWOX, downregulation of prosurvival pERK, prompt increases in Ca^2+^ influx, and disruption of the IκBα/WWOX/ERK prosurvival signaling [54]. WWOX7-11 (AGLDD) peptide does not block ceritinib cytotoxicity in vitro (Figure 6B). In stark contrast, WWOX7-11 is even more powerful than WWOX7-21 in blocking cancer cell growth in vivo [54]. The mechanism by which WWOX7-11 works in an opposite fashion in vitro and in vivo is not clear. Antibody against WWOX286-299 also enhances ceritinib-mediated 4T1 cell sphere explosion and death (Appendix A). The observations indicate that WWOX with S14 phosphorylation is pro-survival for cancer cells [54].

To understand the functional mechanism for WWOX7-21-mediated cancer suppression, we determined that exogenous WWOX7-21 peptide colocalizes with membrane type II TGFβ receptor (TGFβRII) [54]. WWOX antibody or pY33-WWOX antibody pulls down TGFβRII, WWOX, and HYAL-2 in the lipid raft, indicating that TGFβRII is an additional component of the HYAL-2/WWOX/SMAD4 signaling [36]. WWOX7-21 peptide may undergo self-polymerization in vitro [54]. WWOX7-21 peptide binds the first WW domain of WWOX on the cell membrane [36]. Antibody against WWOX7-21 peptide pulls down the full-length WWOX, membrane HYAL-2 and TGFβRII, again further validating that WWOX7-21 peptide is able to initiate the HYAL-2/WWOX/SMAD4 signaling [36].

## 12. Phosphorylation Status of WWOX in the HYAL-2/WWOX/SMAD4 Complex and Disease Progression

Endogenous pS14-WWOX protein is accumulated in the lesions of growing tumors and AD brains [50,51]. Suppression of WWOX phosphorylation at S14 by Zfra4-10 peptide results in significant reduction in cancer growth in mice [51], and enhanced restoration of memory loss and mitigation of AD-like symptoms in triple transgenic (3xTg) mice [50]. pY33-WWOX, but not pS14-WWOX, is needed for binding with HYAL-2 and SMAD4 in vivo [36]. Whether pY33-WWOX is downregulated and pS14-WWOX upregulated in the HYAL-2/WWOX/SMAD4 complex during disease progression is unknown. Zfra, known as zinc finger-like protein that regulates apoptosis, is a 31-amino-acid protein [18,50,51,52,74,75]. As short as 7 amino acids, Zfra4-10 (RRSSSCK) strongly suppresses the progression of cancer and AD [50,51]. S8 is a determined phosphorylation site, and is a key to the function of Zfra [18,50,51]. Additionally, Zfra4-10 blocks NF-κB-mediated inflammation and accelerates degradation of proteins in the AD pathologies, as well as induction of the activation of HYAL-2+ CD3− CD19− lymphocytes to kill cancer in vivo [50,51,52,53].

## 13. Zfra4-10 or WWOX7-21 Activates the HYAL-2/WWOX/SMAD4 Signaling for Z Cell Activation and Suppression of Disease Progression In Vivo

An additional mechanism for the peptide function in vivo is that Zfra peptides, including Zfra4-10 and Zfra1-31, become polymerized in the circulation and then trapped in the spleen as the polymers emit red and green fluorescence [51]. Zfra binds membrane HYAL-2 of spleen Z cells, but not T and B cells. This binding leads to initiation of the HYAL-2/WWOX/SMAD4 signaling that results in activation of memory cytotoxic Z cells [50,51,52,53]. The activated Z cell, which is HYAL-2+ CD3− CD19−, is highly potent in killing cancer cells both in vitro and in vivo. The Zfra-activated Z cells have never encountered cancer cell antigens, and yet they effectively recognize and kill cancer cells [50,51,52,53]. Unlike the generation of chimeric antigen receptor T-cells (CAR-T), Z cell activation does not require pre-exposure to cancer antigens. Polymerized Zfra peptide probably possesses certain motifs or domains similar to those in the cancer antigens. Transfer of activated Z cells to naïve mice or cancer-growing mice confers suppression of cancer growth [50,51,53]. Activated Z cells also kill cancer cells in vitro [51,53]. Additionally, Zfra4-10 peptide is capable of restoring memory loss and inhibiting neurodegeneration in triple transgenic mice for AD [50], suggesting that activated Z cells are involved in memory restoration in the AD mice.

As mentioned above, WWOX7-21 peptide enhances the efficacy of ceritinib-mediated cancer cell death in vitro, and activates Z cells in mice and thereby blocks tumor growth [54]. WWOX7-21 peptide colocalizes with membrane type II TGF-β receptor (TβRII) [36]. Both WWOX7-21 peptide and TβRII simultaneously undergo internalization in response to TGF-β1, suggesting that WWOX7-21 complexes with TβRII. Actually, presence of a HYAL-2/WWOX/TβRII complex is found in the lipid raft of cells in many organs (e.g., liver, spleen and brain) of control mice [36]. The amino acid sequence of WWOX7-21 peptide is located in the *N*-terminal leader sequence and a small part of the first WW domain. WW domain and SDR domain may undergo self-polymerization or hetero-polymerization at the intramolecular or intermolecular level [36]. WWOX7-21 peptide may undergo self-polymerization in phosphate-buffered saline and binds the WW domain area [36]. Binding of WWOX7-21 peptide with membrane HYAL-2 is unknown. Importantly, WWOX7-21 peptide strengthens the binding of WWOX with intracellular proteins for blocking cancer and AD progression. Again, WWOX7-21 peptide antibody pulls down the full-length WWOX, membrane HYAL-2 and TGFβRII, further validating the concept that WWOX7-21 peptide is able to initiate the HYAL-2/WWOX/SMAD4 signaling [36].

## 14. Zfra-Induced Spleen Z Cell Activation Requires De-Phosphorylation at S14, Y33 and Y61 in WWOX In Vivo

When lymphocytic cells are stimulated with calcium ionophore A23187 and phorbol ester (IoP) in vitro, endogenous WWOX undergoes dephosphorylation at Y33 and Y61 and acquires S14 phosphorylation during the course of T cell differentiation (Figure 7A) [56]. When mice receive pS14-WWOX7-21 peptide via tail vein injections, dramatic upregulation of cytotoxic CD8α+ T and CD19+ B cells, but not Foxp3+ T regulatory (Treg) cells, is observed in the spleen [54]. The induced T and B cells fail to kill cancer cells. Although pS14-WWOX7-21 peptide assists melanoma and breast cancer cells to grow even faster and bigger in mice, this peptide may be of therapeutic value in bringing up T/B cells from immunodeficient patients [53]. Without S14 phosphorylation, WWOX7-21 peptide fails to induce T/B cell expansion [54].

The differentiation of Z cells is different from that of T/B cells, in which S14, Y33 and Y61 are dephosphorylated (Figure 6A). When mice receive Zfra4-10 peptide to build anticancer response, Zfra4-10 peptide in circulation becomes polymerized and is trapped in the spleen [51]. The polymerized Zfra4-10 peptide remains in the spleen for at least 2 months. It continuously stimulates the generation of activated Z cells. WWOX undergoes dephosphorylation at S14, Y33 and Y61 [54] (Figure 7B). The activated spleen Z cells relocate to normal organs such as liver and lung, and cancer lesion sites in organs [51]. CD19+ or CD27+ B cells are not involved in Zfra-mediated cancer suppression [54].

## 15. Zfra4-10 or WWOX7-21 Increases the Binding of Endogenous WWOX with Intracellular Protein Partners, Which Contributes to Cancer Growth Suppression In Vivo

Zfra4-10 or WWOX7-21 mediates the signaling of HYAL-2/WWOX/SMAD4 for causing Z cell activation [19,50,51,53]. When activated Z cells encounter cancer cells, the activated Z cells undergo clonal expansion and strongly kill cancer cells in vitro and in vivo [54]. Notably, when mice receive Zfra4-10 or WWOX7-21 peptide alone via tail vein injections once per week for three consecutive weeks followed by subcutaneous cancer inoculations, these mice exhibit an increased binding of endogenous WWOX with ERK, C1qBP, NF-κB, IBA1, p21, CD133, JNK1, COX2, OCT4, and GFAP in the spleen, brain, and lung [19,53]. The stronger the binding, the weaker the cancer growth [53], suggesting that Zfra4-10 or WWOX7-21 peptide-induced HYAL-2/WWOX/SMAD4 signaling is involved in the increased binding of WWOX with its partners.

In stark contrast, when mice receive both Zfra4-10 and WWOX7-21 peptides in combination via tail vein injections, both peptides nullify each other’s function, which leads to enhanced tumor growth [53]. That is, reduced binding of endogenous WWOX with target proteins in organs and tumor lesions occurs, which allows enhanced cancer growth [53]. Both Zfra4-10 and WWOX7-21 peptides tend to undergo aggregation in phosphate-buffered saline [51,53]. We do not exclude the possibility that both Zfra4-10 and WWOX7-21 peptides covalently bind each other and thereby lose their function in signaling and anticancer activity [53].

Indeed, we have shown that de novo synthesized Zfra can covalently conjugate with cellular protein targets [51,53]. The “zfrated proteins” are readily subjected to rapid degradation independently of the proteasome/ubiquitination system [50,51]. By yeast two-hybrid analysis, Zfra physically binds WWOX at the *N*-terminal first WW domain and the *C*-terminal SDR domain [18]. Zfra also suppresses WWOX phosphorylation at Tyr33 and inhibits WWOX-mediated apoptosis [18]. By using a bicistronic pIRES-based vector for expressing Zfra and WWOX constructs, a covalent Zfra/WWOX-DsRed complex (~76 KDa) can be identified in reducing SDS-PAGE [53]. Finally, stimulating the membrane HYAL-2/WWOX complex with HYAL-2 antibody or sonicated hyaluronan (HAson) leads to Z cell activation for killing cancer cells in vivo and in vitro [53] (Figure 6B). Sonication of hyaluronan for 8 hr (HAson8) is needed to maximize the function of HA in suppressing cancer growth in vivo [50]. Overall, Zfra4-10 binds membrane HYAL-2, induces dephosphorylation of WWOX at S14, Y33 and Y61, and drives Z cell activation for the anticancer response (Figure 6B). Thus, Zfra4-10 and WWOX7-21 peptides, HAson8, and HYAL-2 antibodies are of therapeutic potential for cancer suppression.

A critical question is whether activated Z cells retard AD progression. Zfra4-10 and Zfra1-31 peptides are potent in restoring memory loss in triple transgenic 3xTg mice [50,51]. Zfra-mediated suppression of S14 phosphorylation in WWOX (>90%) is needed for Z cell activation, which positively correlates with prevention and blocking of AD progression in 3xTg and *Wwox* heterozygous mice [50]. If activated Z cells are involved in preventing neurodegeneration, then the activated Z cells are likely to secrete cytokines to support neuronal survival. Alternatively, activated Z cells may have acquired capability in traveling through the blood-brain barrier. Indeed, certain populations of inflammatory immune cells are capable of passing through the blood-brain barrier with the assistance of IL17-induced reactive oxygen species (ROS) to damage brain endothelial cells [76].

## 16. Switching the HYAL-2/WWOX/SMAD4 Signaling from Bubbling Cell Death to Membrane Blebbing by Replacing HYAL-2 with p53 

Aberrant signaling may occur as a result of multiplex signaling pathways that accounts for the association of WWOX with diseases [13,14,19,20,21,53,66]. For example, high-molecular-weight hyaluronan (HA) binds cell membrane hyaluronidase HYAL-2 to initiate the HYAL-2/WWOX/SMAD4 signaling [44]. When cancer cells are transiently overexpressed with SMAD4, WWOX and HYAL-2, HA induces cell death [44]. This type of non-apoptotic cell death is designated as bubbling cell death (BCD) [16,17,18,44,56]. BCD starts from the nucleus. No caspase activation, flip over of membrane phosphatidylserine, or DNA fragmentation are involved [16,17,18].

To examine the signaling flow of a trimolecular complex, cells are electroporated with ECFP-SMAD4, EGFP-WWOX and DsRed-p53 constructs for transient overexpression, and then treated with HA of 2 to 4 million Daltons, followed by time-lapse FRET microscopy. Membrane hyaluronidase HYAL-2 binds and degrades HA, and meanwhile induces the formation of a signaling complex of ectopic ECFP-SMAD4, EGFP-WWOX and DsRed-p53 [44,58]. In the initial driving phase (Phase I), there is an increased binding energy of WWOX with both p53 and SMAD4, which lasts 7 h and then jumps to a greater extent of execution force (Phase II) for 11 h. The cells undergo membrane blebbing without apoptotic death [44,58]. When ectopic p53 is replaced by an intracellular form of ectopic HYAL-2(-sp), the signaling of SMAD4/HYAL-2(-sp)/WWOX leads to nuclear-based bubbling cell death. The resulting driving force in phase I is shortened to 3 h and the execution phase to 20 h [58]. Taken together, replacing p53 with HYAL-2 allows the switching of membrane blebbing to bubbling cell death, suggesting the machinery of bubbling cell death emanating from the nucleus is switched on.

## 17. Discussion and Perspectives

In summary, the biological functions of the HYAL-2/WWOX/SMAD4 signaling pathway have been thoroughly described in this perspective article. Both TGF-β1 and hyaluronan are able to bind membrane HYAL-2, and thereby drive the downstream signaling with WWOX and SMAD4 [26]. This pathway controls cell growth, differentiation and death [26], spleen Z cell differentiation and activation [50,51], and traumatic brain injury [44]. During the hyaluronan/HYAL-2 signaling, hyaluronan initiates the ectopic HYAL-2/WWOX/SMAD4 pathway in cells, which results in bubbling cell death [44]. Agonist Zfra4-10 or WWOX7-21 peptide, HYAL-2 antibody or sonicated hyaluronan HAson8, also activate the HYAL-2/WWOX/SMAD4 signaling for spleen Z cell activation in order to kill cancer cells in vitro and in vivo.

WWOX is localized not only in the cell membrane, but also in many subcellular organelles such as the mitochondrion (at outer membrane), lysosome, Golgi complex (with ribosome) and nucleus (with chromosome) in both live and dying cells, as determined by confocal microscopy and immunoelectron microscopy [43,44,77,78,79,80]. Quite frequently, WWOX acts as a downstream adaptor to receive the signal from ligand/receptor interactions. WWOX physically binds tumor suppressor NF2/merlin [78], whereas the biological effect is unknown. Additionally, WWOX binds p73, ERBB-4, SIMPLE, WWBP1, WWBP2, EZRIN, AP-2γ, RUNX-2, DVL-2, TMEM207, AMOTp130, and many others in a PPXY or PPPY-dependent manner [12,24,33,80,81,82,83,84,85,86], indicating its homeostatic role in regulating numerous signaling pathways. WWOX restricts the enzymatic activity of ERK and JNK in a Y33 phosphorylation-dependent manner, and thereby limits enzyme-dependent tau hyperphosphorylation for enhancing the progression of AD. Thus, loss of WWOX may lead to enhanced cancer growth and accelerated AD progression [12,13,21,42].

Whether the HYAL-2/WWOX/SMAD4 signaling limits cancer and neurodegeneration in a balanced manner is unknown. When the WWOX/TIAF1/p53 triad is accumulated in cancer cells, cancer cell growth suppression and death are likely to occur (Figure 8) [55]. This protein triad strongly inhibits cell migration, anchorage-independent transforming growth, and SMAD promoter activation. Notably, when there is a functional antagonism between p53 and WWOX in vivo, this event favors cancer growth and enhanced AD progression. Another interesting finding is that WWOX7-21 or Zfra4-10 peptide increases the strength of endogenous WWOX binding with its protein partners in vivo. The stronger the binding, the better the cancer suppression [53]. Whether this correlates with inhibition of AD progression is being determined.

Outstanding questions that remain to be resolved are: (1) How can we enhance the binding of WWOX with its intracellular protein partners, thereby leading to inhibition of cancer growth? Appropriate peptides or small chemicals will be designed and used to enhance the binding strength of WWOX with its protein partners, so as to suppress cancer growth and meanwhile inhibit AD progression. Indeed, either Zfra4-10 or WWOX7-21 peptides increase the binding strength of WWOX with its partners [53]. (2) How can activated Z cells exert cytotoxicity for leading to cancer cell death? Is this via cytokines or direct physical contact with cancer cells? Z cells need to be primed for activation by Zfra or WWOX peptides or sonicated hyaluronan via the signaling of HYAL-2/WWOX/SMAD4 [53]. Thus, utilization of either Zfra4-10 or WWOX7-21 peptide for Z cell activation in cancer treatment is feasible and is being tested in animals. 3) Can HAson8 provide a better strength in Z cell activation for cancer treatment? HAson8 is generated by sonicating hyaluronan at a sufficient kilohertz, timing and temperature [53]. Although all the designed drugs utilize the HYAL-2/WWOX/SMAD4 signaling to control cancer growth, it is necessary to compare the efficacy of HAson8, Zfra4-10, WWOX7-21 and other available drugs in treating cancer in vivo.

## Figures and Tables

**Figure 1 cells-11-02137-f001:**
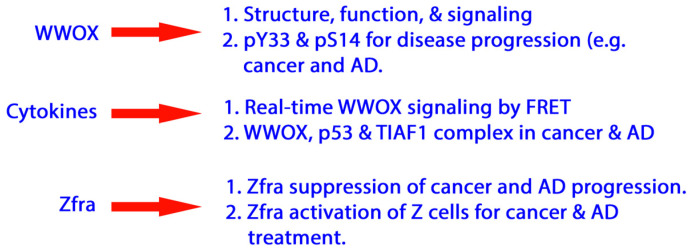
A diagram for the presentation scheme in this article.

**Figure 2 cells-11-02137-f002:**
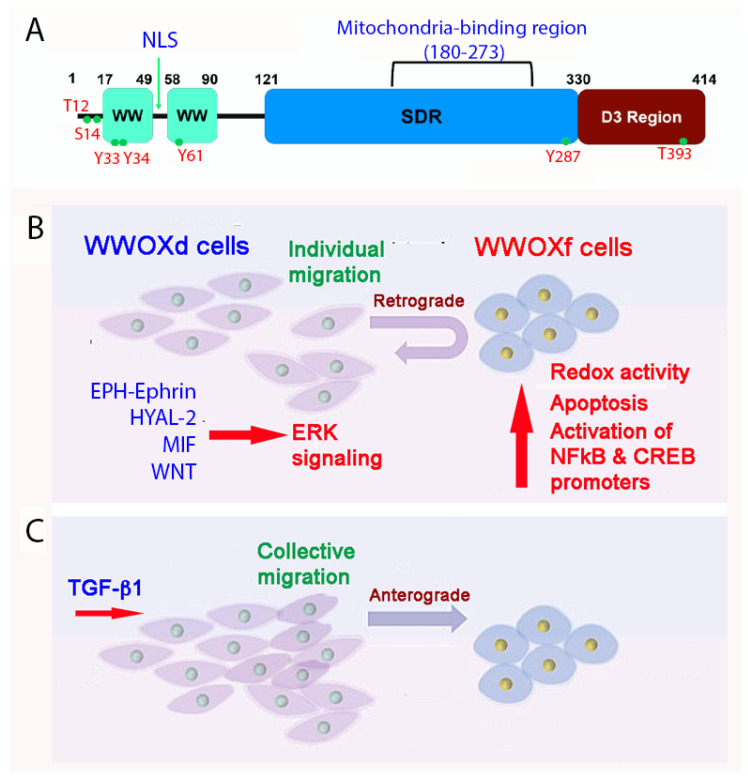
A schematic model for how WWOXd cells undergo retrograde migration upon facing WWOXf cells. (**A**) The primary structure of WWOX is depicted. (**B**) When WWOXd cells face WWOXf cells, WWOXd cells undergo activation of multiple signaling pathways, including MIF, HYAL-2, WNT and EPH-Ephrin, that converge to MEK/ERK pathway leading to retrograde migration. WWOXd cells have no physical contacts with WWOXf cells but are able to kill WWOXf cells from a distance. (**C**) TGF-β abolishes the retrograde migration and apoptosis, and induces anterograde migration and merger of WWOXd and WWOXf cells. Similarly, antibodies against MIF, HYAL-2 or EPH-Ephrin, and WNT inhibitors abolish the retrograde migration of WWOXd cells in facing WWOXf cells [36,37].

**Figure 3 cells-11-02137-f003:**
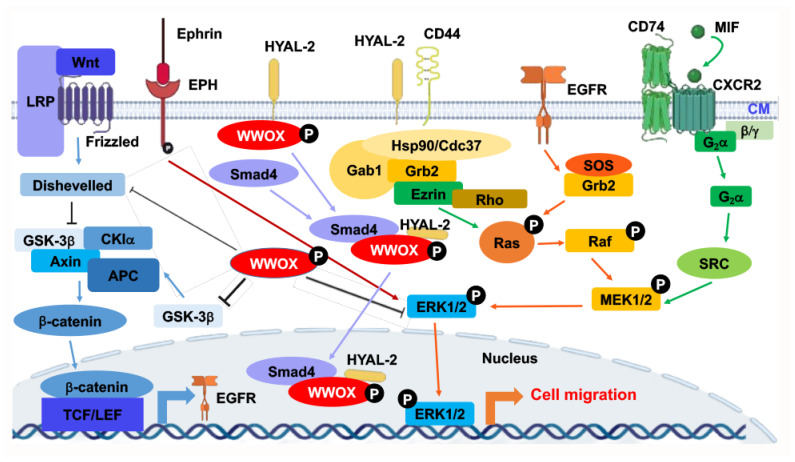
Signal pathways converged to ERK for controlling retrograde migration of WWOXd cells upon facing WWOXf cells. Pathways, including WNT, HYAL-2, Ephrin/EPH, EGF/EFGR, and MIF, allow signaling to drive down to ERK1/2. ERK1/2 controls the retrograde migration of WWOXd cells upon facing WWOXf cells. Suppression of WNT, MIF, HYAL-2, or Ephrin/EPH causes the retrograde migration of WWOXd cells, anterograde migration and finally merger with WWOXf cells [36,37]. The role of HYAL-2/CD44 in controlling retrograde migration remains to be established.

**Figure 4 cells-11-02137-f004:**
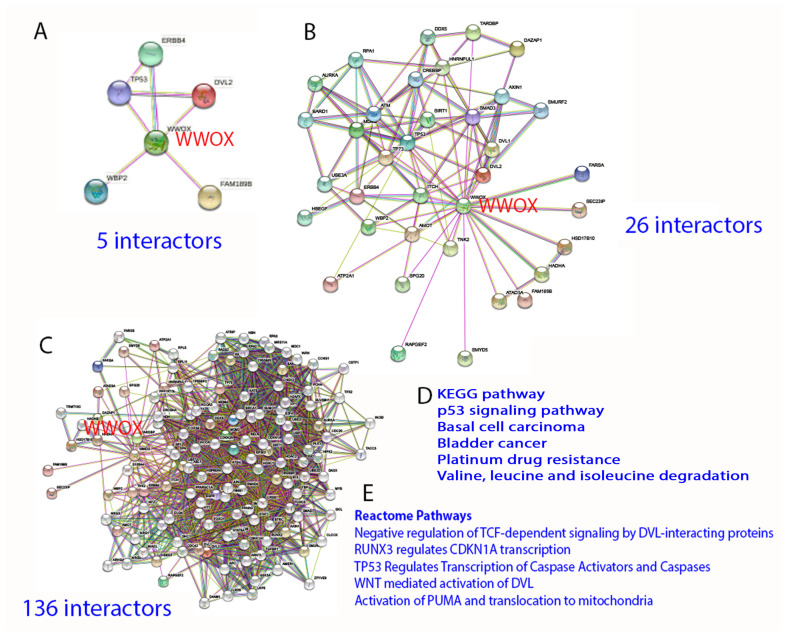
STRING analysis of WWOX signaling network. (**A**–**E**) The analysis starts with 5 interactors, and is then expanded to 26 and 136 interactors, respectively. Representative KEGG and Reactome pathways are shown.

**Figure 5 cells-11-02137-f005:**
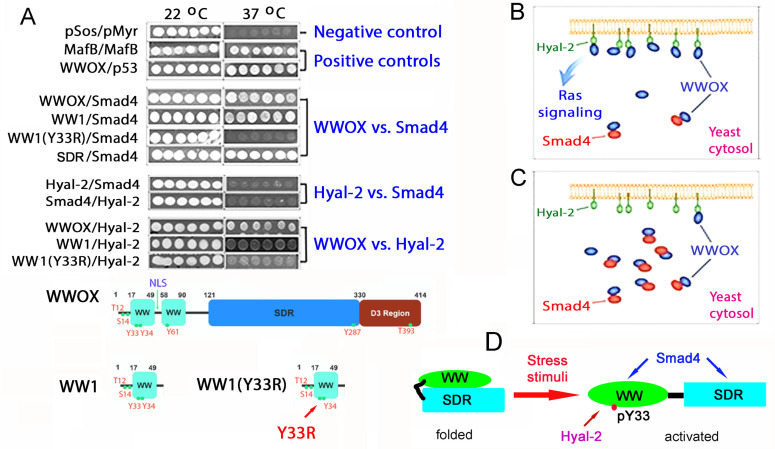
Binding of cytosolic WWOX by SMAD4 reduces WWOX interaction with membrane-bound HYAL-2 and thereby inhibits yeast cell growth at 37 °C. (**A**) By Ras rescue-based yeast two-hybrid analysis [11,25,26,38,48,70,71,72]. Binding of WWOX with p53 or MafB self-interaction allows the growth of yeast at 37 °C using a selective agarose plate containing galactose. No yeast growth at 37 °C is seen for the empty pSos and pMyr vectors. SMAD4 binds the *N*-terminal first WW domain of WWOX (WW1) and the *C*-terminal SDR domain binds SMAD4 or HYAL-2 [44]. HYAL-2 binds WW1 in a pY33-dependent manner [44]. WW1(Y33R) fails to bind SMAD4 or HYAL-2. HYAL-2 and SMAD4 fail to bind each other [44]. (**B**) In competitive binding assay, ectopic HYAL-2 (target; green) is designed for anchoring onto the cell membrane of yeast. Various amounts of cytosolic SMAD4 (competitor; red) are used to bind a constant amount of cytosolic WWOX (bait; dark blue) and reduce WWOX binding to membrane HYAL-2. (**C**) When an increased amount of SMAD4 is overly expressed, SMAD4 binds WWOX to prevent the signaling for HYAL-2/WWOX and thereby leads to growth suppression of yeast. (**D**) When WWOX undergoes Y33 phosphorylation, it becomes unfolded to let its binding with HYAL-2 and SMAD4. WW domain physically interacts with SDR domain in non-activated WWOX [36,37]. (data adapted from Reference [44] with major modifications and new interpretations; permission not required from Oncotarget).

**Figure 6 cells-11-02137-f006:**
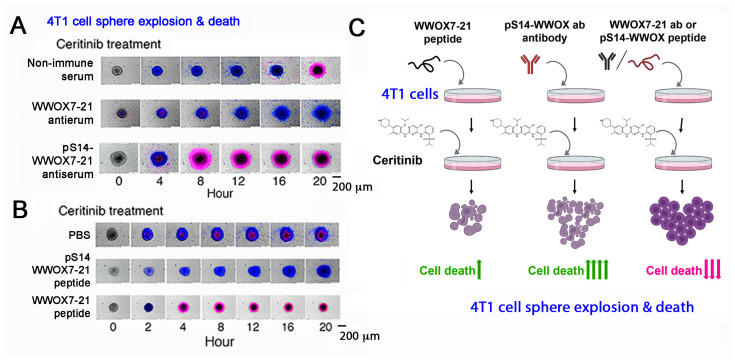
pS14-WWOX7-21 peptide and WWOX7-21 antibody block ceritinib-mediated breast 4T1 cell sphere explosion and death. Cells are pretreated with an indicated peptide (10 μM) or antiserum (1:100 dilution) for 30 min and then treated with ceritinib (30 μM) for time-lapse microscopy. Nuclear stains DAPI (blue) and Propidium Iodide (red) are added in the culture. (**A**,**B**) Both WWOX7-21 peptide and pS14-WWOX7-21 antibody enhance 4T1 cell sphere explosion and death. pS14-WWOX7-21 peptide strongly blocks ceritinib-mediated 4T1 sphere explosion and cell death. (**C**) Summary of enhancers and inhibitors for ceritinib-mediated cell death. Upward green arrows indicate increases in cell death, and downward red arrows for reduced cell death. (image data adapted from Reference [54].

**Figure 7 cells-11-02137-f007:**
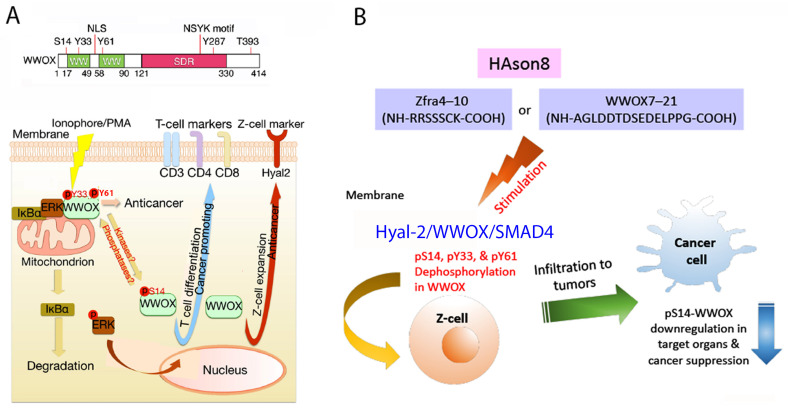
HYAL-2/WWOX/SMAD4 signaling for Z cell activation to block cancer growth and AD progression. (**A**) Ionophore/PMA (IoP) induces T cell differentiation by dephosphorylating pY33 and pY61 in the WWOX. However, S14 phosphorylation is needed for T cell differentiation [56]. Whether Ionophore/PMA induces Z cell differentiation is unknown. Zfra peptide (e.g., Zfra4-10) induces Z cell activation via dephosphorylation of pS14, pY33 and pY61 in WWOX. Membrane HYAL-2 is rapidly upregulated in Z cells [54]. (**B**) Agonists, including Zfra4-10, WWOX7-21, WWOX7-11 and 8 hr-sonicated hyaluronan (HAson8), induce Z cell activation via HYAL-2/WWOX/SMAD4 signaling for blocking cancer growth and retarding AD progression [54].

**Figure 8 cells-11-02137-f008:**
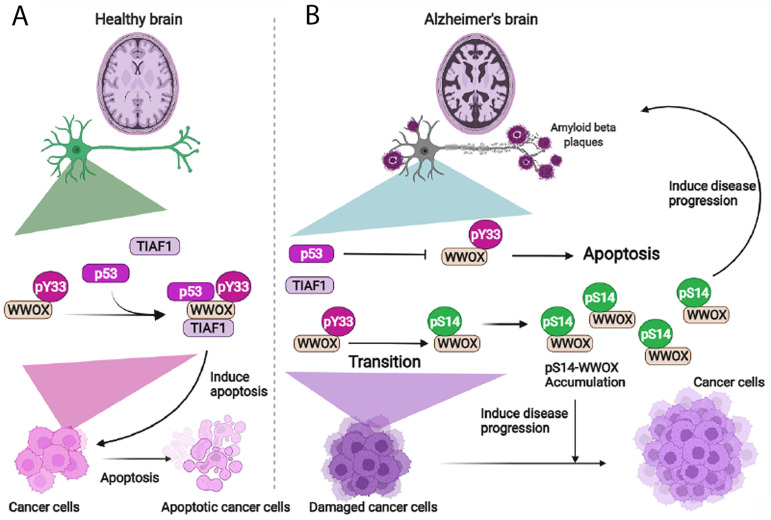
WWOX signals with p53 and TIAF1 for cancer suppression. (**A**) Overexpressed triad p53/WWOX/TIAF1 protein complex induces apoptosis of cancer cells. WWOX in the complex is Y33 phosphorylated. The effect of the triad on normal brain cells is unknown. (**B**) pY33-WWOX can be converted to pS14-WWOX. pS14-WWOX is gradually accumulated in the lesions of cancer and AD brain during disease progression. pS14-WWOX supports cancer growth and enhances neurodegeneration such as in AD. Zfra4-10 peptide suppresses phosphorylation of WWOX at S14 and thereby restores memory loss in AD mice and inhibits cancer growth [50,51].

## Data Availability

Not applicable.

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
