# Peer review of "WWOX Controls Cell Survival, Immune Response and Disease Progression by pY33 to pS14 Transition to Alternate Signaling Partners"

_cells, 2022, doi:10.3390/cells11142137_

Round 1
Reviewer 1 Report
In this review, Tsung-Yun Liu and co-workers describe the influence of different complexes involving the tumor suppressor WWOX and the phosphorylation state of this protein, in cancer and mental illness, particularly in Alzheimer's disease (AD).
Mainly, they expose various observations and experimental arguments indicating that the action of WWOX on cancer and mental diseases is dependent of its phosphorylation state, indeed, pY33-WWOX inhibits whereas pS14-WWOX promotes cancer and AD. WWOX complexed with TIAF1 could inhibit mental disorders by preventing the aggregation of TIAF1 induced notably by TGFbeta1. TGFbeta1, Hyluronan and Zfra induce the formation of the Hyal-2/WWOX/Smad4 complex. This complex would inhibit cancer by activating Z cells in the spleen, which are highly potent in killing cancer cells. The phosphorylation of WWOX on its tyrosine 33 is indispensable for its interaction with TIAF1, Hyal-2 and Smad4. They also indicate that the peptides, Zfra4-10, WWOX7-21 and sonicated Hyaluronan induce Z cell activation via Hyal-2/WWOX/Smad4 complex, suggesting that these peptides could be used as therapeutic agents in cancer.
This review is interesting and provide numerous information. I noticed some points that need to be considered.- Line 211, the authors declare that the two tumor suppressors “p53 and WWOX may counteract functionally with each other, and thereby lead to cancer growth enhancement and development of AD pathologies in vivo”. However, they indicate that pY33-WWOX interacts with p53 to induce cell death (lines 277 and 278). It has recently been suggested that WWOX acts as a tumor suppressor by maintaining the stability of p53 in breast cancer (Abdeen et al. Cell Death and Disease (2018)9:832). Therefore, WWOX and p53 would rather cooperate in cancer and AD instead of mutually inhibiting each other. Authors should discuss this apparent contradiction.
- Lines 337 and 338, in the title WWOX7-221 is mentioned but not in the following text.
- Figure 3A does not fully correspond to the text. Lines 357 and 358, "When lymphocyte cells are stimulated...WWOX undergoes dephosphorylation at Y33 and Y61..." WWOX dephosphorylation at Y61 is not visible in the diagram. Lines 388 and 389, "Hyal-2/WWOX/Smad4 signaling for Z cell activation to block cancer growth and AD progression" the Hyal-2/WWOX/Smad4 complex is not indicated in the diagram, instead the IkBα complex /ERK/WWOX is indicated, as well as IkBα degradation?
Author Response
Dear Reviewer 1:
We appreciate your outstanding efforts. Your comments are very helpful. We have answered point-by-point to your comments as shown below:
(A) Line 211, the authors declare that the two tumor suppressors “p53 and WWOX may counteract functionally with each other, and thereby lead to cancer growth enhancement and development of AD pathologies in vivo”. However, they indicate that pY33-WWOX interacts with p53 to induce cell death (lines 277 and 278). It has recently been suggested that WWOX acts as a tumor suppressor by maintaining the stability of p53 in breast cancer (Abdeen et al. Cell Death and Disease (2018)9:832). Therefore, WWOX and p53 would rather cooperate in cancer and AD instead of mutually inhibiting each other. Authors should discuss this apparent contradiction.
Answer: Thank you for the challenging question. Binding of p53 with WWOX was first reported in 2001, then 2003 and 2005 [references #11, 24, 59]. These binding studies were done by co-immunoprecipitation and by yeast two hybrid analysis. Stabilization of p53 by WWOX was first reported in 2005 [59]. The observations were then reproduced by Abdeen et al. in 2018 (Cell Death and Disease (2018)9:832). In the 2005 study, yeast two hybrid analysis was utilized to map the physical binding and stabilization of pS46-p53 by proapoptotic pY33-WWOX. The pY33-WWOX/pS46-p53 complex induces apoptosis many cell lines in vitro [59]. Whether the complex induces cancer cell death in vivo is unknown. We suspect that when phosphorylation of WWOX in Y33 is downregulated and S14 upregulated, the binding strength between WWOX and p53 is reduced. pS14-WWOX is critical in promoting cancer growth and enhancing the progression of Alzheimer’s disease (AD) [40, 41]. Thus, we predict that when pY33-WWOX is shifted to pS14-WWOX, the binding strength of pS14-WWOX with pS46-p53 is reduced, and this facilitates the growth of cancer and the progression of Alzheimer’s disease [40, 41]. By the same token, binding of pS14-WWOX with other intracellular proteins is weakened and thereby favors the growth of cancer and the progression of AD [19, 43]. Together, when WWOX is downregulated or Y33 phosphorylation is blocked, UV-induced p53 protein expression is suppressed, even though p53 mRNA levels are stable [59]. Thus, the proteomic profiles of intracellular protein-binding partners of pS14- and pY33-WWOX, respectively, remains to be further investigated.
(B) Lines 337 and 338, in the title WWOX7-21 is mentioned but not in the following text.
Answer: Thank you. As requested, we added “WWOX7-21 peptide enhances the efficacy of ceritinib-mediated cancer cell death in vitro, and activates Z cells in mice and thereby blocks tumor growth [44]. WWOX7-21 peptide colocalizes with membrane type II TGF-ß receptor (TßRII) [33]. Both WWOX7-21 peptide and TßRII simultaneously undergo internalization in response to TGF-ß1, suggesting that WWOX7-21 complexes with TßRII. Actually, presence of a HYAL-2/WWOX/TbRII complex is found in the lipid raft of cells in many organs (e.g. liver, spleen and brain) of control mice [33]. The amino acid sequence of WWOX7-21 peptide is located in the N-terminal leader sequence and a small part of the first WW domain. WW domain and SDR domain may undergo self-polymerization or hetero-polymerization at the intramolecular or intermolecular level [33]. WWOX7-21 peptide may undergo self-polymerization in phosphate-buffered saline and binds the WW domain area [33]. Binding of WWOX7-21 peptide with membrane HYAL-2 is unknown. Importantly, WWOX7-21 peptide strengthens the binding of WWOX with intracellular proteins for blocking cancer and AD progression. Again, WWOX7-21 peptide antibody pulls down the full-length WWOX, membrane Hyal-2 and TGFßRII, further validating WWOX7-21 peptide is able to initiate the Hyal-2/WWOX/Smad4 signaling [33].”
(C) Figure 3A does not fully correspond to the text. Lines 357 and 358, "When lymphocyte cells are stimulated...WWOX undergoes dephosphorylation at Y33 and Y61..." WWOX dephosphorylation at Y61 is not visible in the diagram. Lines 388 and 389, "Hyal-2/WWOX/Smad4 signaling for Z cell activation to block cancer growth and AD progression" the Hyal-2/WWOX/Smad4 complex is not indicated in the diagram, instead the IkBα complex /ERK/WWOX is indicated, as well as IkBα degradation?
Answer: Thank you. Again, our apologies. Regarding “Lines 357 and 358”, we have revised the original Figure 3A and added pY61 (now Figure 6A). Regarding “Lines 388 and 389”, we have added Hyal-2/WWOX/Smad4 complex to the revised Figure 6B. Please also note that the Hyal-2/WWOX/Smad4 signaling complex is shown in the new Figure 2.
(D) Key additional changes
At the request of other reviewers, we have made additions and revisions as follows:
1) New Figure 2: A detailed graph regarding signal pathways converged to ERK for controlling retrograde migration of WWOXd cells upon facing WWOXf cells.
2) New Figure 3: This graph deals with WWOX signaling network and functional implication. The connection was done with STRING analysis [https://string-db.org/cgi/network?taskId=brQHTnXWILzL&sessionId=b73ctYMesgwG ]. The computational generated frist level of WWOX-binding protein network reveals its connection with 5 interactor proteins. We randomly showed the binding network for 26 and 136 interactors. Indeed, the network can be expanded tremendously . The significance of this approach allows demonstration of the complexity in the biological network and yet still yielding a functional machinery. The critical issue is that domain/domain or phosphorylation-dependent networking cannot be effectively generated. Also, the computational approach still fails to prove our observations that the stronger the binding of WWOX with intracellular proteins, the better the suppression of cancer growth and retardation of the progression of Alzheimer’s disease [43]. The binding affinities among many known and unknown proteins are not well defined.
3) A new Figure 4 is shown, which shows “binding of cytosolic WWOX by Smad4 reduces WWOX interaction with membrane-bound Hyal-2 and thereby inhibits yeast cell growth at 37oC.” This graph provides a clear illustration of how yeast two-hybrid works and the suitability in measuring protein domain/domain interactions.
Reviewer 2 Report
The pleiotropic function of WWOX oxidoreductase has attracted the attention of many researchers over the last decade, especially in relation to cancer progression and neurological disorders. Therefore, the topic selected for review is of interest but in its present form is difficult to follow and, in my opinion, would benefit greatly if the authors could introduce changes to facilitate its understanding.
For instance:
Please write in the introduction a clear description of the content of the review and why the review focuses on some functional aspects and not others. The main idea of what you want to tell in this review is missing, the reader jumps from paragraph to paragraph without knowing what to expect in the next paragraph.
I find it strange that some paragraphs start with "we have used (line 150)" as if the review focused only on the authors' work.
I missed in Figure 1 the schematic representation of the domain composition of the protein, indicating the position of the phosphorylation sites. I also think that a picture of the domains or a link to structures or models available should be included at the beginning of the text.
I also believe that adding appropriate schematic representations of the interactions described in each section will greatly assist the reader. For example, in section 8: The dynamics of WWOX/TIAF1/p53 triad formation and the functional antagonism between p53 and WWOX in enhancing cancer progression and Alzheimer's disease. Many proteins are mentioned. It is almost impossible for the lay reader to follow the signaling pathway and which protein binds to which protein.
Minor comments
The authors should also try to be consistent with the nomenclature (Smad/SMAD). The PPxY motif includes the PPPY sequence.
Author Response
Dear Reviewer 2:
We appreciate your outstanding effort. Your comments are very helpful. We have answered your questions point-by-point as follows:
(A) The pleiotropic function of WWOX oxidoreductase has attracted the attention of many researchers over the last decade, especially in relation to cancer progression and neurological disorders. Therefore, the topic selected for review is of interest but in its present form is difficult to follow and, in my opinion, would benefit greatly if the authors could introduce changes to facilitate its understanding.
Answer: Thank you. We agree.
(B) For instance:
Please write in the introduction a clear description of the content of the review and why the review focuses on some functional aspects and not others. The main idea of what you want to tell in this review is missing, the reader jumps from paragraph to paragraph without knowing what to expect in the next paragraph.
Answer: Thank you for the great point. As requested, in the Introduction, we have now clarified the content of the review step by step and explained how and why the review focuses are. A illustrated scheme is now added in the text, which shows the flow of how this article is written.
(C) I find it strange that some paragraphs start with "we have used (line 150)" as if the review focused only on the authors' work.
Answer: As requested, "we have used“ has been deleted. The sentence has been rephrased as “Förster resonance energy transfer (FRET) microscopy is a feasible approach to measure WWOX signaling and associated functions”.
(D) I missed in Figure 1 the schematic representation of the domain composition of the protein, indicating the position of the phosphorylation sites. I also think that a picture of the domains or a link to structures or models available should be included at the beginning of the text.
Answer: As requested, we have added a schematic graph for the primary structure of WWOX, which contains two N-terminal WW domains, C-terminal SDR domain, and phosphorylation sites (new Figure 1).
(E) I also believe that adding appropriate schematic representations of the interactions described in each section will greatly assist the reader. For example, in section 8: The dynamics of WWOX/TIAF1/p53 triad formation and the functional antagonism between p53 and WWOX in enhancing cancer progression and Alzheimer's disease. Many proteins are mentioned. It is almost impossible for the lay reader to follow the signaling pathway and which protein binds to which protein.
Answer: Thank you for the great point. Please note that the story of WWOX/TIAF1/p53 triad was originally shown in the last figure (Figure 4) of the original manuscript (now Figure 7). As requested, we have added few selective graphs in the text: i) A scheme for the flow of this article (line 89 to 90), ii) New Figure 2 is a detailed signal pathways for converging to ERK that regulates the retrograde migration of WWOXd cells upon facing WWOXf cells, iii) New Figure 3 is a STRING analysis of WWOX signaling network, and iv) New Figure 4 illustrates how domain/domain interactions work in yeast two-hybrid analysis and how competitive binding assay was established, and molecular details provided.
(F) Minor comments
The authors should also try to be consistent with the nomenclature (Smad/SMAD). The PPxY motif includes the PPPY sequence.
Answer: Again, thank you. We have fixed every needed word. Please refer to those marked in red. That is, based upon international nomenclature standard, the names of proteins are set in upper cases. Also, we fixed "PPXY (e.g. PPPY) motif".
(G) Key additional changes
At the request of other reviewers, we have made additions and revisions as follows:
1) New Figure 2: A detailed graph regarding signal pathways converged to ERK for controlling retrograde migration of WWOXd cells upon facing WWOXf cells.
2) New Figure 3: This graph deals with WWOX signaling network and functional implication. The connection was done with STRING analysis [https://string-db.org/cgi/network?taskId=brQHTnXWILzL&sessionId=b73ctYMesgwG ]. The computational generated frist level of WWOX-binding protein network reveals its connection with 5 interactor proteins. We randomly showed the binding network for 26 and 136 interactors. Indeed, the network can be expanded tremendously . The significance of this approach allows demonstration of the complexity in the biological network and yet still yielding a functional machinery. The critical issue is that domain/domain or phosphorylation-dependent networking cannot be effectively generated. Also, the computational approach still fails to prove our observations that the stronger the binding of WWOX with intracellular proteins, the better the suppression of cancer growth and retardation of the progression of Alzheimer’s disease [43]. The binding affinities among many known and unknown proteins are not well defined.
3) A new Figure 4 is shown, which shows “binding of cytosolic WWOX by Smad4 reduces WWOX interaction with membrane-bound Hyal-2 and thereby inhibits yeast cell growth at 37oC.” This graph provides a clear illustration of how yeast two-hybrid works and the suitability in measuring protein domain/domain interactions.
Reviewer 3 Report
The review manuscript entitled "WWOX: A Fragile Tumor Suppressor with Pleotropic Functions" by Liu et al. is an interesting and comprehensive overview of WWOX and its interaction with other proteins and signaling pathways. The review is well written and organized and provides the reader with the latest information on this tumor suppressor gene/protein.
However, I would suggest a scheme of the gene structure and figures explaining the interactions with other proteins and pathways.
Minor remark, there are 12 known isoforms of p53.
Author Response
Dear Reviewer 3:
We appreciate your outstanding effort. Your comments are very helpful. We have answered your questions point-by-point as follows:
(A) The review manuscript entitled "WWOX: A Fragile Tumor Suppressor with Pleotropic Functions" by Liu et al. is an interesting and comprehensive overview of WWOX and its interaction with other proteins and signaling pathways. The review is well written and organized and provides the reader with the latest information on this tumor suppressor gene/protein.
Answer: Thank you.
(B) However, I would suggest a scheme of the gene structure and figures explaining the interactions with other proteins and pathways.
Answer: As per request from you and other reviewers, we have added the primary amino acid structure of WWOX, which shows the presence of WW domains, SDR domain, phosphorylation sites, and others (new Figure 1). In the new Figure 2, we have detailed the signaling pathways which are converged to ERK for controlling the retrograde migration of WWOXd cells upon facing WWOXf cells. As requested, in the new Figure 3, by STRING analysis, we showed the WWOX-connected signaling network.
(C) Minor remark, there are 12 known isoforms of p53.
Answer: Thank you. Our apologies. Yes, there are 12 isoforms.
(D)
At the request of other reviewers, we have made additions and revisions as follows:
1) New Figure 2: A detailed graph regarding signal pathways converged to ERK for controlling retrograde migration of WWOXd cells upon facing WWOXf cells.
2) This is answer to your request. New Figure 3: This graph deals with WWOX signaling network and functional implication. The connection was done with STRING analysis [https://string-db.org/cgi/network?taskId=brQHTnXWILzL&sessionId=b73ctYMesgwG ]. The computational generated frist level of WWOX-binding protein network reveals its connection with 5 interactor proteins. We randomly showed the binding network for 26 and 136 interactors. Indeed, the network can be expanded tremendously . The significance of this approach allows demonstration of the complexity in the biological network and yet still yielding a functional machinery. The critical issue is that domain/domain or phosphorylation-dependent networking cannot be effectively generated. Also, the computational approach still fails to prove our observations that the stronger the binding of WWOX with intracellular proteins, the better the suppression of cancer growth and retardation of the progression of Alzheimer’s disease [43]. The binding affinities among many known and unknown proteins are not well defined.
3) A new Figure 4 is shown, which shows “binding of cytosolic WWOX by Smad4 reduces WWOX interaction with membrane-bound Hyal-2 and thereby inhibits yeast cell growth at 37oC.” This graph provides a clear illustration of how yeast two-hybrid works and the suitability in measuring protein domain/domain interactions.
Round 2
Reviewer 2 Report
The authors have answered all comments satisfactory. Thank you.
Author Response
Dear Reviewer:
Thank you again.
Nanshan Chang